# CRMP/UNC-33 organizes microtubule bundles for KIF5-mediated mitochondrial distribution to axon

Ying-Chun Chen, Hao-Ru Huang, Chia-Hao Hsu, Chan-Yen Ou*

Institute of Biochemistry and Molecular Biology, College of Medicine, National Taiwan University, Taipei, Taiwan

* chanyen@ntu.edu.tw

## Abstract

Neurons are highly specialized cells with polarized cellular processes and subcellular domains. As vital organelles for neuronal functions, mitochondria are distributed by microtubule-based transport systems. Although the essential components of mitochondrial transport including motors and cargo adaptors are identified, it is less clear how mitochondrial distribution among somato-dendritic and axonal compartment is regulated. Here, we systematically study mitochondrial motors, including four kinesins, KIF5, KIF17, KIF1, KLP-6, and dynein, and transport regulators in *C. elegans* PVD neurons. Among all these motors, we found that mitochondrial export from soma to neurites is mainly mediated by KIF5/UNC-116. Interestingly, UNC-116 is especially important for axonal mitochondria, while dynein removes mitochondria from all plus-end dendrites and the axon. We surprisingly found one mitochondrial transport regulator for minus-end dendritic compartment, TRAK-1, and two mitochondrial transport regulators for axonal compartment, CRMP/UNC-33 and JIP3/UNC-16. While JIP3/UNC-16 suppresses axonal mitochondria, CRMP/UNC-33 is critical for axonal mitochondria; nearly no axonal mitochondria present in *unc-33* mutants. We showed that UNC-33 is essential for organizing the population of UNC-116-associated microtubule bundles, which are tracks for mitochondrial trafficking. Disarrangement of these tracks impedes mitochondrial transport to the axon. In summary, we identified a compartment-specific transport regulation of mitochondria by UNC-33 through organizing microtubule tracks for different kinesin motors other than microtubule polarity.

## Author summary

Functional and structural distinct axonal and dendritic compartments demand specific regulation of mitochondrial distribution. While most of previous studies examined mitochondrial transport in cultured neurons or at a certain segment of axon, we genetically ablated five motors and four regulators and study their roles in a sensory neuron that has two microtubule polarity distinct dendrites and an axon. We showed that dynein regulates mitochondrial distribution in neurites based on microtubule polarity regardless of axon or dendrite, while KIF5, but not the other kinesin motors, mediates an axon-specific

**Data Availability Statement:** All relevant data are within the manuscript and its Supporting Information files.

**Funding:** This study was funded by grants: MOST 106-2311-B-002-034, MOST 107-2311-B-002-

006, and Jade Picking projects NTU-JP-109L7206, NTU-JP-109L7231 from Ministry of Science and Technology (MOST), Taiwan, and National Taiwan University to C-YO. The funders had no role in study design, data collection and analysis, decision to publish, or preparation of the manuscript.

**Competing interests:** The authors have declared that no competing interests exist.

mitochondrial transport. We surprisingly found that CRMP/UNC-33 is critical for axonal mitochondria and it supports a stable microtubule population to form a spindle-like structure in neuronal cell body as tracks for trafficking. These results indicate that CRMP/UNC-33 promotes high-order organization of microtubule bundles in neurons, which are cells without a clear centralized microtubule organization center but still possess highly organized microtubule bundles. As defects of mitochondrial trafficking are implicated in most of the neurodegeneration diseases, this study manifests specific roles of motors and regulators in distinct axonal and dendritic compartments, and it further reveals how the microtubule network forms in the neuronal cell body.

## Introduction

Neurons are polarized cells with functionally distinct cellular compartments including the somatodendritic and axonal compartments. In the somatodendritic compartment, branched dendrites cover receptor fields for receiving signals, while typically a long and sheer axon reaches target region in order to output signals. Apart from most other cells, the cellular processes of neurons are particularly long. For example, the axonal length of some motor neurons can be more than one meter long in mammals [1]. Hence, neuronal functions demand a sound cellular transport system for distributing structural components, signaling molecules, and vital organelles. Mitochondria play multiple critical roles for neuron metabolism including the regulation of cell apoptosis, calcium ion reservoir, and energy supply. Mitochondria are recruited to the sites with high energy demands such as the growth cones, neurite branching points, synapses, and injury loci [2–5]. Defects of mitochondrial transport in neuron are implicated in many neurodegenerative diseases, including Huntington's disease, amyotrophic lateral sclerosis (ALS) and Alzheimer's disease [6,7]. Thus, it is important to understand the mechanism of proper mitochondrial transport and distribution in neurons.

In neurons, mitochondria are mainly transported by the microtubule (MT)-based transport system [8]. Axonal MTs are uniformly aligned with their plus-end toward the axonal terminal, while the orientations of MTs in dendrites are mixed with both plus-end-out and minus-end-out MTs in mammalian culture neurons [9]. Mitochondria are attached to molecular motors via adaptor proteins for efficient movement. The minus-end motor cytoplasmic dynein and the plus-end motor KIF5 drive the retrograde and anterograde transport respectively by tethering to mitochondria via adaptor Milton/TRAK1 and Miro on mitochondrial outer membrane [10–14]. Although the essential components for mitochondrial transport in axon have been identified, it remains unclear how mitochondrial distribution in other neuronal compartments such as dendrite is regulated. To identify how mitochondrial transport to a specific compartment is regulated, we examined potential regulators involved in KIF5 and dynein-mediated transport, including TRAK1/Milton, JIP3/UNC-16, and CRMP/UNC-33. The JNK kinase-interacting protein JIP3/UNC-16 acts as a potential motor regulator that binds kinesin-1 subunits and the dynein light intermediate chain [15–17]. Collapsin response mediator proteins (CRMPs) are a family of cytosolic phosphoproteins involved in the establishment of neuronal polarity and axon growth [18,19]. CRMP can potentially promote microtubule assembly by binding to tubulin heterodimer and stabilize microtubule *in vitro* or in cultured cells [18,20,21]. Interestingly, CRMPs are also shown to regulate the transport of kinesin-1 complex and dynein [18,22]. However, whether CRMPs regulate the transport of mitochondria in neuron remains unclear.

The regulation of mitochondrial transport is likely different in structurally and functionally distinct subcellular compartments such as soma, axon, and dendrites. Compartment-specific cargos can be carried by different motors, which selectively adopt distinct MT tracks according to MT post-translational modification and MT-associated proteins. For example, a microtubule-associated septin (SEPT9) and microtubule-associated protein 2 (MAP2) differentially hinder KIF5-mediated cargo transport, while promote or allow KIF1 transport [23,24]. Also, KIF5 prefers to move over acetylated MT bundles, while KIF1 selectively interacts with tyrosinated MT bundles [25]. However, it is unclear what the regulatory mechanism is to organize MT bundles with different posttranslational modification and how MT organization regulates mitochondrial transport in neuron.

In order to study how mitochondrial distribution is regulated in different neuronal compartments *in vivo*, we used elaborately branched mechanosensory PVD neurons in *Caenorhabditis elegans* as our model system. PVDL and PVDR (left and right side) locate beneath the worm epidermis and cover most of the body for sensing harsh touch and cold temperature. A PVD neuron has an easily distinguishable axon (with plus-end-out MTs) innervating the ventral nerve cord and two highly branched menorah-like dendrites including an anterior primary dendrite (with minus-end-out MTs) extending from the cell body toward the head and a posterior primary dendrite (with plus-end-out MTs) toward the tail [26,27]. In this study, we systemically characterized 5 motors implicated in mitochondrial transport including the kinesin-1 family motor KIF5/UNC-116, the kinesin-2 family KIF17/OSM-3, two kinesin-3 family motors KIF1/UNC-104 and KLP-6, and the cytoplasmic dynein DHC-1 for manifesting their specific roles in different neuronal subcellular compartments. Next, to identify compartmental specific regulatory mechanism for axonal or dendritic mitochondrial distribution, we examined motor regulators, including TRAK1/Milton, JIP3/UNC-16, and CRMP/UNC-33. We identified KIF5/UNC-116 as the main motor responsible for axonal mitochondria, which are strictly regulated by regulators; while JIP3/UNC-16 suppresses axonal mitochondria, CRMP2/UNC-33 is essential for mitochondrial axonal distribution. We showed that KIF5/UNC-116 is associated with a specific sub-population of MT bundles, and these MT bundles are organized by CRMP/UNC-33.

## Result

### Mitochondrial distribution is regulated by motor regulator MIRO-1 in PVD neurons

The microtubule polarity in PVD neurons has been well characterized in previous studies by tracing microtubule plus ends [26,27]. While axonal MTs are uniformly plus-end-out, it was shown that MTs are oriented as minus-end-out direction in the anterior dendrite, and plus-end-out in the posterior dendrite. In order to study mitochondrial distribution and its regulation mechanism, we generated a transgenic line *ntuIs1* to label PVD cell membrane by myristoylated mCherry and mitochondria by TOMM20::GFP (Fig 1A and 1B). In wild-type worms, the average number of mitochondria in all neurites (the axon and anterior and posterior dendrites) of a PVD neuron was 39.9 ± 1.5 (Fig 1F). To describe the mitochondrial distribution, the anterior dendrite is subdivided into three sections (-1, -2, and -3) from the proximal to the distal end, and the posterior dendrite is defined as section (+1), and the axon as section (+A) (Fig 1A). Mitochondria were about evenly distributed in these dendritic sections (Fig 1B and 1G). When we genetically ablated the mitochondrial motor receptor MIRO-1 by introducing the frameshift allele *tm1966* that disrupts the mitochondrial transmembrane domain, mitochondrial distributions in all sections of dendrites and the axon were severely abolished (2.4 ± 0.4) (Fig 1C and 1F), suggesting that mitochondrial transport in all PVD neurites

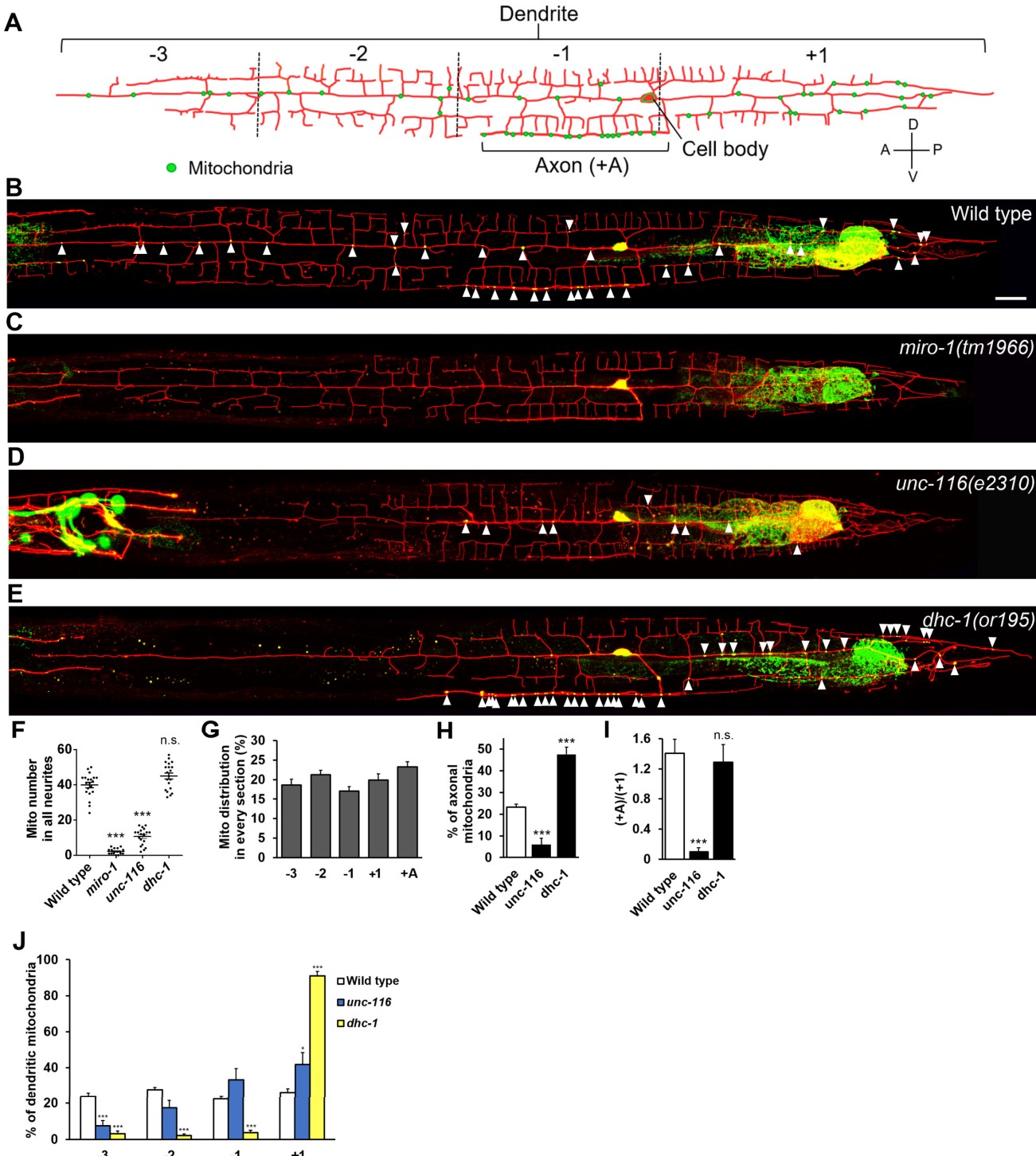

**Fig 1. Mitochondrial distribution is regulated by motor-based transport system in PVD neurons.** (A) Diagram of highly-branched PVD neuron morphology and mitochondria. PVD is composed of MT plus-end-out oriented posterior dendritic (+1) and axonal sections (+A) and a MT minus-end-out anterior dendritic section. The anterior dendritic region is subdivided into three sections, (-1) to (-3) from proximal to distal. Mitochondria (green dots) distribute evenly through primary dendrites and the axon. (B) Wild-type PVD neuron expressing myr-mCherry and mitochondrial marker TOMM-20::GFP under *Pdes-2* promoter (*ntuIs1*). Mitochondria are indicated by arrowheads. All worm images were oriented with the anterior toward left and the ventral toward down. Scale bar, 20 μm. (C-E) *miro-1(tm1966)* (C), *unc-116(e2310)* (D), *dhc-1(or195)* (E) mutant PVD neurons. (F) Quantification of the number of mitochondria in all neurites in wild type and indicated mutants. One-way ANOVA, compared to wild type. (G) Percentage of mitochondrial number at each PVD section in wild-type worms. (H) Percentage of axonal mitochondria. χ2 test, compared to wild type. (I) The ratio of axonal to posterior dendritic

mitochondrial number. χ2 test. (J) Distribution of mitochondria in dendritic sections. χ2 test. Error bar represents SEM. n.s., not significant, $^*$P<0.05, $^{**}$P<0.01, $^{***}$p<0.001. n = 20.

requires MIRO-1-mediated motor systems. Meanwhile, mitochondria accumulated in the PVD cell body (S1A and S1B Fig). Thus, these results suggest that both dendritic and axonal mitochondrial distribution require MIRO-1-mediated transport machinery in PVD neurons.

### Distinctive roles of kinesin and dynein motors in mitochondrial transport in the PVD axon and dendrites

To understand how different motors regulate mitochondria distribution in neuronal compartments, we examined four different kinesins and dynein known to be involved in neuronal mitochondrial transport, including KIF5/UNC-116, KIF17/OSM-3, KIF1/UNC-104, kinesin-3/KLP-6, and cytoplasmic dynein/DHC-1. KIF5 plays important roles in bi-directional mitochondrial transport in mice motor neuron, and dynein is required for retrograde transport in *Drosophila* motor axon [12,28]. KIF1Bα is required for mitochondrial transport [29], and KLP-6 is a putative orthologue of kinesin-3, which regulates *C. elegans* mitochondrial morphology in body wall muscles and mouse neuronal cells [30].

In *unc-116* loss-of-function mutant animals, the number of total mitochondria in all neurites was significantly reduced (10.7 ± 1.0) compared to wild type (39.9 ± 1.5, Fig 1D and 1F and S1 Table), while mitochondrial intensity in the cell body was increased (S1A and S1B Fig), indicating that KIF5/UNC-116 is responsible for mitochondrial export from the cell body to neurites.

Of these remaining neurite mitochondria in *unc-116* mutant, only 6.0% can distribute to the axon compared to 23.2% in wild-type animals (Fig 1H). In contrast, in *dhc-1(or195)* mutant, we observed that most of the neurite mitochondria distribute in the +1 dendritic section (48.6%) and the axon (47.6%), while very few mitochondria were left in the anterior dendrite (Fig 1E, 1H and 1J). While microtubules are in plus-end-out orientation in both axonal (+A) and the posterior dendritic (+1) compartments, the ratio of mitochondria in axon to the posterior dendrite remained the same in *dhc-1* mutant as wild type (Fig 1I). However, in the *unc-116* mutant, the ratio was greatly reduced to 0.1, compared to 1.4 in wild type (Fig 1I). Although UNC-116 regulates the polarity of dendritic MTs in DA9 neuron [31], the polarity of PVD MT remains plus-end-out in posterior dendrite and the axon [27]. Based on these analyses, we conclude that KIF5/UNC-116 has a more specific role in mitochondrial axonal transport, but dynein mediates minus-end-directed transport regardless of axon or dendrite.

We further investigated the role of KIF17/OSM-3, KIF1/UNC-104, and KLP-6 in mitochondrial distribution. In both *unc-104(e1265)* and *osm-3(p802)* mutant animals, the number of total mitochondria in neurites was not affected, but the percentage of axonal mitochondria in both mutants was unexpectedly increased (S1D and S1E Fig and S1 Table), suggesting that OSM-3 and UNC-104 mildly suppresses mitochondrial axonal distribution. In the *klp-6(my8)* loss-of-function mutant, no obvious mitochondrial distribution defects could be found in axons and dendrites (S1F Fig). However, the mitochondrial network formed two aggregated apparatus close to the entry sites of primary dendrites in soma, distinctive to the smooth circular mitochondrial network in wild type (S1G Fig). Overall, we conclude that KIF5/UNC-116 is the main kinesin that mediates mitochondrial transport into neurites from the cell body, while KIF17/OSM-3 and KIF1/UNC-104 play minor roles in regulating axonal distribution, and KLP-6 maintains mitochondrial network in the cell body with no obvious role in neurite mitochondria.

## Motor regulator UNC-16 suppresses axonal mitochondrial distribution while UNC-33 is essential for axonal mitochondria

Since different motors have distinct roles in mitochondrial transport in different subcellular compartments, we hypothesized that there are motor regulators specific for axonal or dendritic mitochondrial distribution. We examined regulators involved in KIF5- and dynein-mediated transport, including TRAK1/Milton, JIP3/UNC-16, and CRMP/UNC-33. TRAK1/Milton binds to both kinesin-1 and dynein and functions as a mitochondrial cargo adaptor [13,32]. In the *trak-1* null mutant *tm1572*, the number of total mitochondria in neurites was significantly reduced by 42.1% (23.1 ± 1.2) compared to wild type (39.9 ± 1.5, Fig 2D), while there was nearly no mitochondrial distribution in anterior dendrites compared to wild type (Fig 2A). However, unexpectedly, mitochondria in posterior dendritic and axonal compartments did not reduce but mildly increased (Fig 2A), indicating that TRAK-1 is specifically responsible for mitochondrial distribution in the MT minus-end-out anterior dendrite but not the MT plus-end-out axon or posterior dendrite.

We further investigated the role of JNK kinase-interacting protein JIP3/UNC-16/Sunday driver, since it potentially regulates kinesin and dynein-mediated transport [15–17]. The total mitochondrial number in neurites significantly increased by 43.4% in *unc-16* mutant (57.2 ± 2.2) compared to wild type (39.9 ± 1.5) (Fig 2B and 2D). The mitochondrial distribution was biased toward the axon and the posterior dendrite with a 2.7-fold increase in the axonal mitochondria and a 1.9-fold increase in the posterior dendritic mitochondria (S1 Table), while the anterior dendrite contains fewer percentages of mitochondria, with a significant reduction in the -3 section (Fig 2G). These results support the idea that UNC-16 functions as a gear shifter for switching to minus-end transport. Notably, the ratio of axonal to posterior dendritic mitochondria significantly increased in the *unc-16* mutant, revealing a specific role of UNC-16 in gating axonal transport rather than merely promoting minus-end transport (Fig 2F).

In contrast to UNC-16, we found that CRMP/UNC-33 acts in an opposite role in controlling mitochondrial distribution. Total neurite mitochondria were reduced to 38% of normal level in *unc-33*(*mn407*) null mutant animals (15.1 ± 1.0) (Fig 2D). While only a few mitochondria were present in the most distal section (-3) of the anterior dendrite, the number of proximal dendritic mitochondria (-1 and +1) were less affected (Fig 2G). Remarkably, we found that almost no mitochondria were present in the axon (0.3 ± 0.2) (Fig 2C and 2E and S1 Table). This strong phenotype indicates that UNC-33 plays a critical role in mitochondrial dispatch to the axon. In addition, we found that *unc-33* mutant has similar mitochondrial distribution defects in mechanosensory neurons. For example, the neuron ALM, which extends a long anterior process with a ventral branch that forms synapses with command neurons at the nerve ring (S2A-A' Fig). In *unc-33* mutant, the number of mitochondria in anterior neurite was significantly reduced (3.5 ± 0.9) compared to wild type (10.5 ± 0.6, S2B and S2C Fig), and mitochondria were absent in the ventral nerve ring branch (indicated by brackets in S2B' Fig). Also, mitochondrial number was strongly reduced in PDE axons in *unc-33* mutant (S2D and S2E Fig).

To sum it up, these results show that individual regulator can specifically regulate mitochondria distribution in different neuronal compartments. We found that TRAK-1 is required for mitochondria in the anterior dendrite, while axonal mitochondria are strictly regulated by UNC-16 and UNC-33.

## UNC-33L rescues dendritic morphology and mitochondrial transport in PVD neuron

In *C. elegans*, *unc-33* encodes three isoforms: UNC-33L, UNC-33M and UNC-33S [33]. With different N-terminal domains, they share common C-terminal domains including the tubulin-

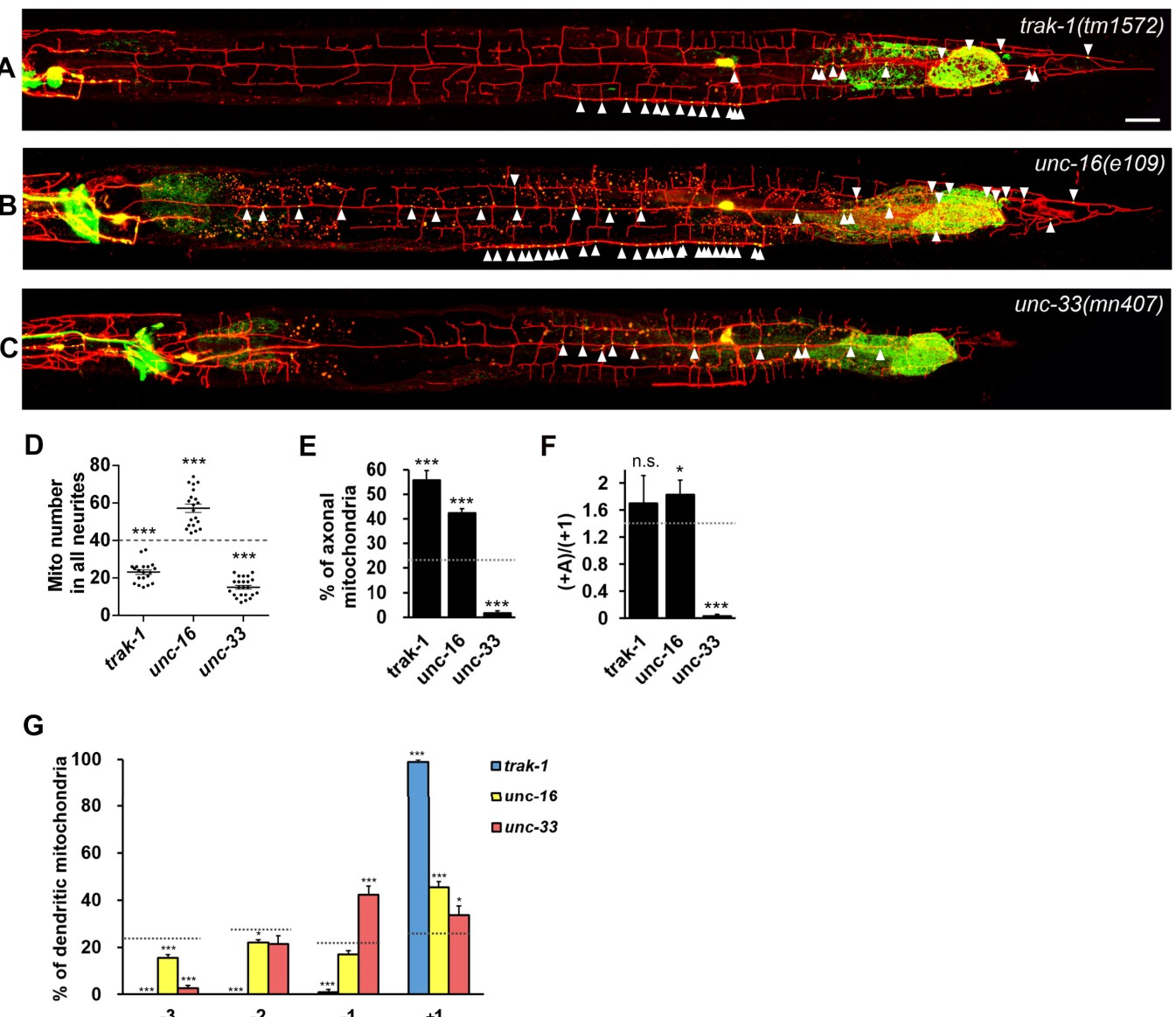

**Fig 2. UNC-33 is essential to axonal mitochondrial distribution.** (A-C) PVD neurons expressing myr-mCherry (red) and mitochondrial marker TOMM-20::GFP (green) (*ntuIs1*) of mitochondrial transport regulator mutants, *trak-1(tm1572)* (A), *unc-16(e109)* (B), and *unc-33(mn407)* (C). Mitochondria are indicated by arrowheads. Scale bar, 20 μm. (D) Quantification of the number of mitochondria in all neurites. One-way ANOVA. (E) Percentage of axonal mitochondria. χ2 test. (F) The ratio of axonal to posterior dendritic mitochondrial number. χ2 test. (G) Mitochondrial distribution in dendritic sections. χ2 test. The dashed line in (D) to (G) indicates the numerical value of wild type. Error bar represents SEM. n.s., not significant, *P<0.05, **P<0.01, ***p<0.001, compared to wild type, n = 20 *(trak-1* and *unc-16)*, n = 23 *(unc-33)*.

and microtubule-binding domains and interaction sites with motors [20–22,34] (S3A Fig). The *mn407* deletion allele we had analyzed disrupts all three isoforms [33]. To test which specific isoform is required for mitochondrial transport, we generated transgenes for these three isoforms driven by the PVD-specific promoter *Pdes-2* (S3A Fig). Only the UNC-33L transgene rescued *mn407* mutant phenotypes, including the decreased dendritic and axonal mitochondrial distribution (S3B and S3C Fig). These data suggest that UNC-33L acts cell-autonomously in the PVD neuron to regulate dendritic and axonal mitochondrial transport.

## UNC-33 acts independently to dynein and KIF5/UNC-116 in regulating axonal transport of mitochondria

Two possible mechanisms could be the basis for how UNC-33 regulates axonal transport of mitochondria in the PVD neuron. First, by inhibiting dynein-dependent transport, UNC-33 promotes mitochondrial transport into the axon. When UNC-33 activity is absent, mitochondria are removed from the axon by the activated dynein complex. Second, UNC-33 is essential for KIF5/UNC-116 activity. Thus, mitochondria could not be transported into the axon by KIF5/UNC-116 when UNC-33 is depleted.

To test the first hypothesis, we examined mitochondrial distribution in the *dhc-1; unc-33* double mutant. If the absence of axonal mitochondria in *unc-33* mutant was caused by active dynein-mediated retrograde transport, the *dhc-1* mutation could reverse the phenotype of *unc-33* mutant. However, the loss of *dhc-1* only restored some axonal mitochondria caused by the loss of *unc-33* (16.6 ± 3.4% in *dhc-1; unc-33* and 1.6±0.0% in *unc-33*), and the percentage of axonal mitochondria was far less than wild type (23.2±1.3%) or *dhc-1* alone (47.6±3.3%) (Fig 3A and 3E). Also, the total mitochondrial number was only slightly increased in *dhc-1; unc-33* (19.8 ± 2.0 in *dhc-1; unc-33* and 15.1 ± 1.0 in *unc-33*) and far less than wild type and *dhc-1* (Fig 3D), which means that UNC-33 unlikely only regulates mitochondrial transport through the dynein motor.

Phenotypes of *unc-116* were very similar to those of *unc-33*. The number of total neurite mitochondria was severely reduced in *unc-116* mutant (73%) and in *unc-33* mutant (62%) compared to wild type. Their mitochondrial distribution both showed stronger reduction in the axon and distal parts of dendrites than proximal regions, which might support the second hypothesis. To test it, we genetically ablated both *unc-33* and *unc-116* in the same animals and expected that the double mutant phenotype would resemble that of single mutants. Surprisingly, we found that total neurite mitochondrial number in *unc-116; unc-33* was further reduced 62% compared to *unc-33* single mutant and 46% compared to *unc-116* single mutant (Fig 3B and 3D), suggesting an additive effect. The data showed that UNC-116 and UNC-33 might function independently to regulate mitochondrial transport out of the cell body. Consistent to this idea, polarized mitochondrial distribution in the posterior dendrite was further reduced in *unc-116; unc-33* (Fig 3G). Taken together, these data suggest that the function of UNC-33 is not completely in the same pathway as dynein and UNC-116.

## *unc-33* mutant disrupts motor polarized distribution and vesicular transport

The above results suggest that UNC-33 regulates mitochondrial transport through pathways not merely modulating of dynein and kinesin activity. We further examined two possibilities, the regulation of the main neuronal cytoskeletal components or mitochondrial distribution through the transport regulator UNC-16. Given that total neurite mitochondria and the ratio of axonal mitochondria both greatly increased in *unc-16* mutant, which is opposite to the phenotype of *unc-33* mutant (Fig 2B, 2D and 2E), we generated *unc-33; unc-16* double mutant to perform an epistatic analysis. If UNC-33 regulates mitochondrial distribution through UNC-16, we should expect an *unc-16*-like phenotype in *unc-33; unc-16* double mutant. Interestingly, we found that the accumulation of mitochondria in the axon and posterior dendrites was almost abolished in *unc-33; unc-16* double mutant animals (Fig 3C and 3E). Since these results showed that *unc-33* acts epistatically to *unc-16*, we inferred that the UNC-16-regulated plus-end transport depends on UNC-33.

We further observed the distribution of UNC-116::GFP to check whether motor transport system functions well under *unc-33* mutant. In wild type, UNC-116::GFP in PVD accumulated and formed puncta in posterior dendrites and the axon (arrowheads in Fig 4A). The KIF5/

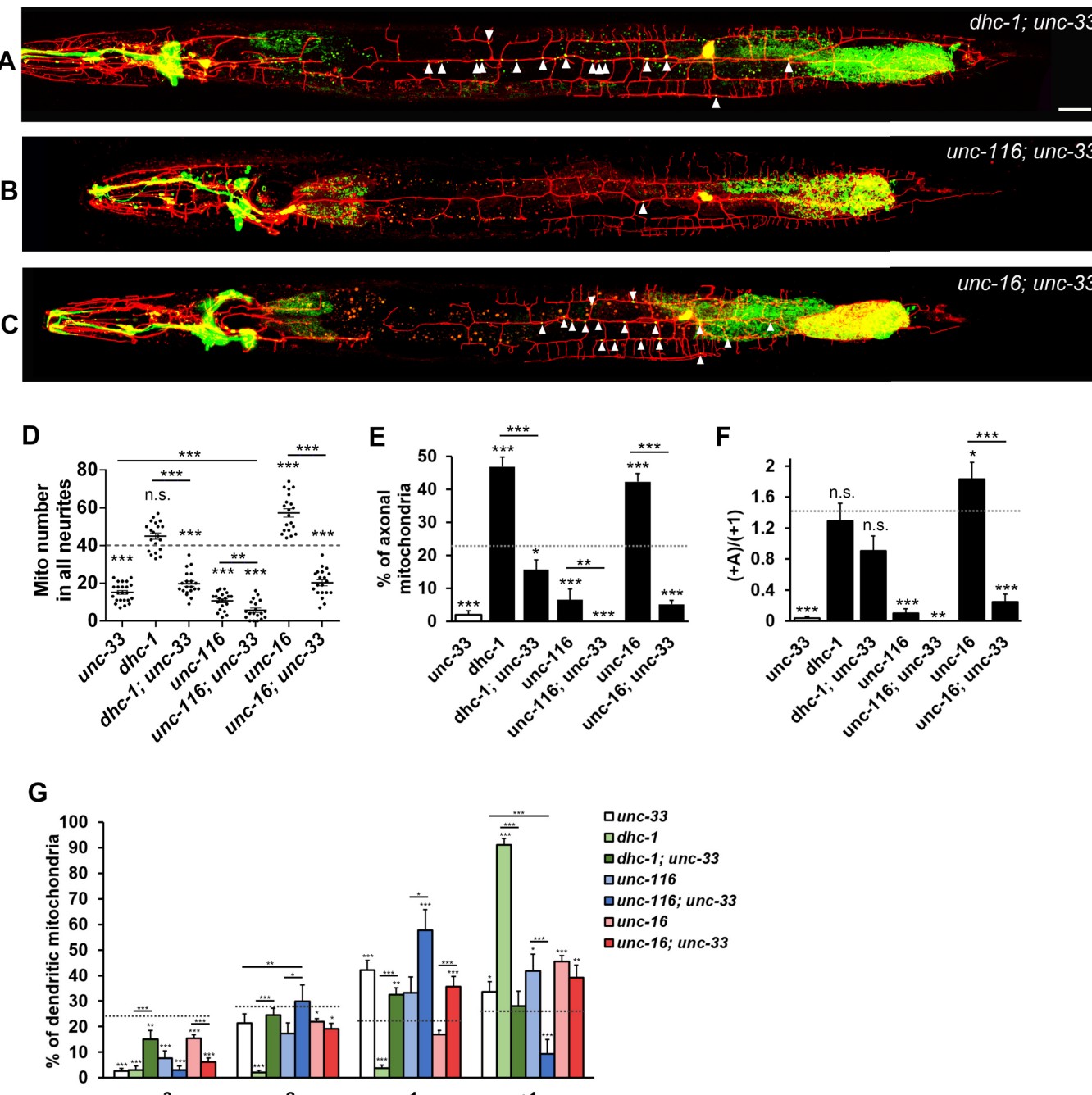

**Fig 3. UNC-33 functions in a *dhc-1*-independent mechanism to regulate mitochondrial transport.** (A-C) PVD neurons expressing myr-mCherry and TOMM-20::GFP (*ntuIs1*) in double mutants, *dhc-1; unc-33* (A), *unc-116; unc-33* (B), and *unc-16; unc-33* (C). Mitochondria are indicated by arrowheads. Scale bar, 20 μm. (D) Quantification of the number of mitochondria in all neurites. One-way ANOVA. (E) Percentage of axonal mitochondria. χ2 test. (F) The ratio of axonal to posterior dendritic mitochondrial number. χ2 test. (G) Mitochondrial distribution in dendritic sections. χ2 test. The dashed line in (D) to (G) indicates the numerical value of wild type. Error bar represents SEM. n.s., not significant, *P<0.05, **P<0.01, ***p<0.001, compared to wild type.

UNC-116 intensity was strongly increased at axon in *unc-16* mutant ([Fig 4B and 4D]). In *unc-33* mutant, however, UNC-116::GFP was no longer enriched at the end of posterior dendrites and the axon, but mislocalized at random dendritic spots ([Fig 4C and 4D]), indicating that

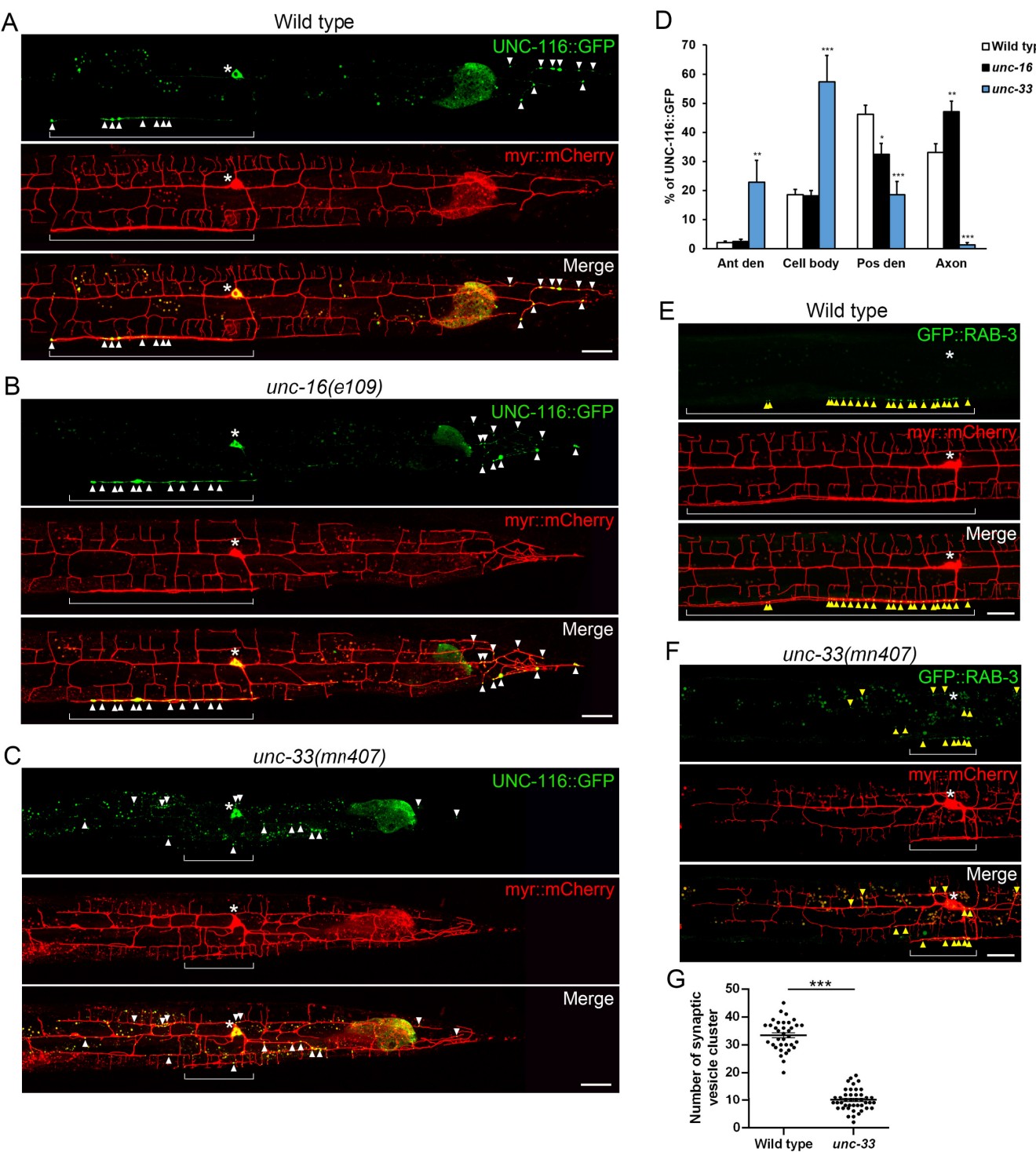

**Fig 4. KIF5/UNC-116 motor polarized distribution and synaptic transport are disrupted in *unc-33* mutant.** (A-C) PVD neurons expressing UNC-116::GFP and myr::mCherry (*ntuIs2*) of wild type (A), *unc-16(e109)* (B) and *unc-33(mn407)* (C). Arrowheads indicate UNC-116::GFP signal and brackets indicate the axon. Scale bar, 20 μm. (D) Percentage of UNC-116::GFP intensity in different neuronal sections. Error bar represents SEM, Student's T-test, *P<0.05, **P<0.01, ***p<0.001, compared to wild type. n = 10. (E and F) Images of PVD neurons expressing synaptic vesicle marker GFP::RAB-3 and myr-mCherry (*wyEx5216*) in wild type (E) and *unc-33(mn407)* (F). Arrowheads indicate synaptic vesicle clusters. (G) Quantification of synaptic vesicle cluster number. Error bar represents SEM, T-test, ***p<0.001, n = 35 (wild type), n = 43 *(unc-33)*.

UNC-33 regulates the polarized pattern of UNC-116::GFP distribution. To investigate whether UNC-33 regulates cargoes of other motors, we traced synaptic vesicles which are transported by KIF1/UNC-104, by labeling synaptic vesicle marker RAB-3 with GFP. In wild type, synaptic vesicles formed clusters in the axons (Fig 4E). In unc-33 mutant, the distribution of synaptic clusters in the axon decreased 66.7%, and showed mislocalized distribution to dendrites (Fig 4F and 4G). These data suggest that UNC-33 might regulate general neuronal cytoskeleton arrangement to orient motor polarized distribution and vesicular transport.

## UNC-33 organizes microtubule bundles in PVD cell body and stabilizes microtubules in the axon

Post-translational modification of microtubule directs motor to axon initial segments, and selectively regulates motor entrance in different neuronal compartments [35]. KIF5 preferentially binds to acetylated stable MTs, while KIF1 preferentially binds to tyrosinated dynamic MTs [25]. We labeled tyrosinated MTs by UNC-104-GFP with rigor mutations (UNC-104-rigor::GFP) that abolished motor activity, and acetylated, stable MTs by UNC-116-mCherry with rigor mutations (UNC-116-rigor::mCherry). We found that UNC-104-rigor::GFP and UNC-116-rigor::mCherry labeled distinct pattern of MTs in the wild-type cell body and neurites (left panel in Fig 5A). UNC-104-rigor::GFP labeled filamentous MTs at the cell body and neurites with a mesh-like pattern and enriched around the nucleus, while UNC-116-rigor:: mCherry labeled thick bundles of MTs that formed spindle-like tracks at the rim of the cell body and extended to neurites (green and red signal, respectively in Fig 5A).

In the absence of UNC-33, the spindle-like UNC-116-rigor signal was severely diminished in the cell body, and the general level of UNC-104-rigor signal in the cortical region of the cell body was also decreased, but the circular mesh of UNC-104-rigor signal was still present around the nucleus (Fig 5A and 5B). Moreover, UNC-116-rigor signal revealed that MTs bundles reduced (Fig 5C) and the remaining MT bundles dispersed into more distal dendritic regions in unc-33 mutant (yellow arrowheads, middle panel, Fig 5A). Quantification of MT intensity along anterior primary dendrite showed that the peak of UNC-116-associated MT signals was at 8.7 ± 0.6 μm from the cell body in wild type, but it was significantly increased to 26.7 ± 2.5 μm in unc-33 mutant in anterior primary dendrite (Fig 5D and 5E). These results indicate that UNC-33 pulls microtubule bundles together around the cell body. Notably, in unc-33 mutant, the intensity ratio of UNC-116-rigor::mCherry (stable MTs) to UNC-104-rigor::GFP (dynamic MTs) was dramatically reduced compared to wild type in the axon, but not in dendrites (Fig 5F), indicating that UNC-33 selectively stabilizes MTs in the axon. The defects of MT organization in unc-33 mutant was rescued by expressing UNC-33L transgene in PVD. Previously we showed that mitochondrial distribution was significantly reduced in ALM anterior neurite and nerve ring branch in unc-33 mutant (S2A–S2C Fig). Hence, we tested if MT structures in touch receptor neurons were also affected in unc-33 mutant. We observed that UNC-116-rigor signal was absent in ALM and PLM cell body, and MT bundles reduced in proximal neurites (S4A and S4B Fig). These data reveal that UNC-33 sticks MT bundles together to form a spindle-like high-order structure in the cell body and selectively maintains stabilized MTs for KIF5.

To examine the subcellular localization of UNC-33, we expressed UNC-33L with GFP inserted between the globular domain and the C-terminal tail (UNC33L::internalGFP) with UNC-116-rigor::mCherry [36]. We found that UNC-33L::internalGFP formed a punctate pattern closely associated with microtubule bundles labelled by UNC-116-rigor::mCherry (Fig 5G). We further labeled F-actin network by tagRFP::UtrCH with UNC-33L::internalGFP. We found that UNC-33 puncta located well on the F-actin bundles (Fig 5H). The localization of

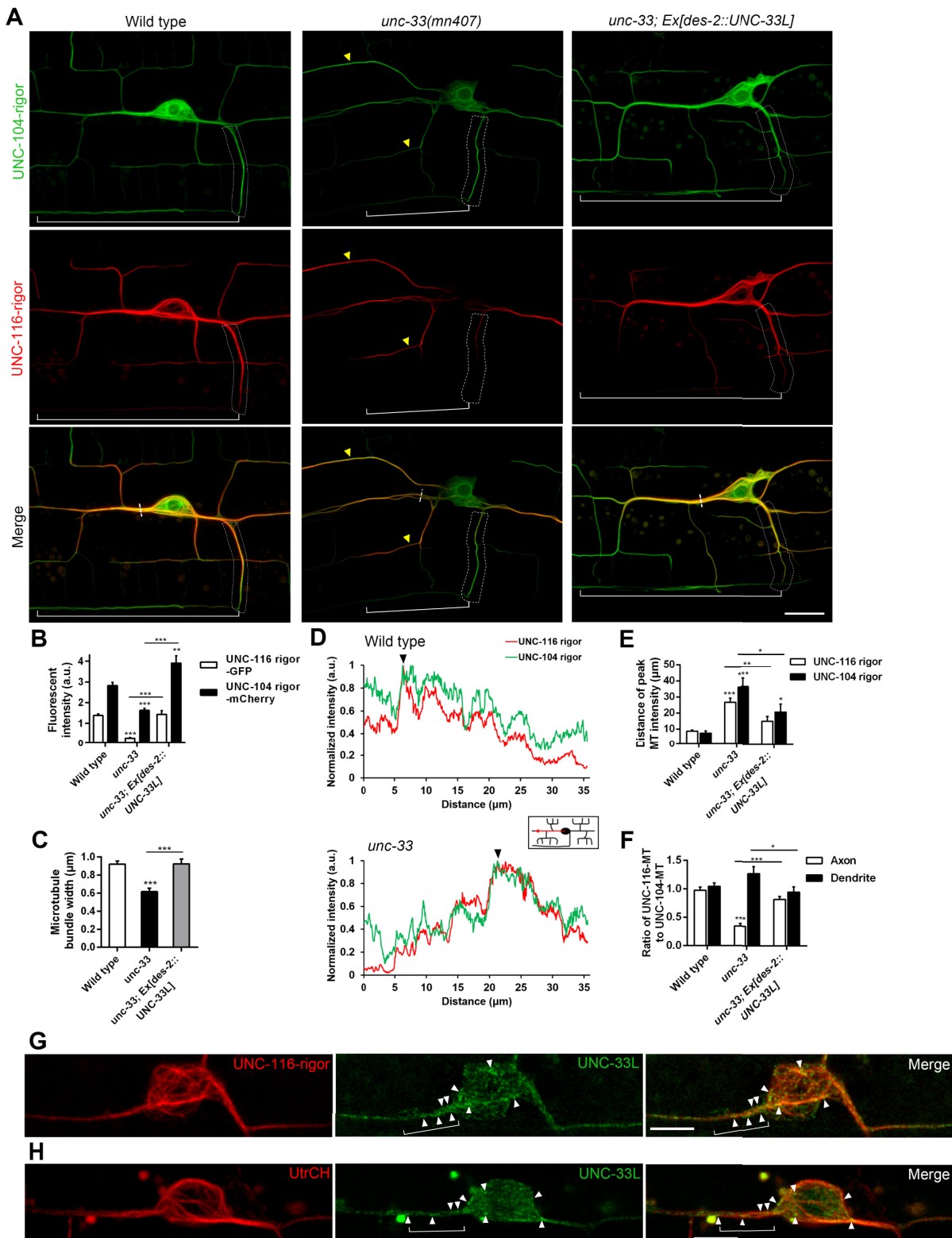

**Fig 5. UNC-33 regulates microtubule organization in the cell body and is required for axonal KIF5/UNC-116-associated microtubule tracks.**
(A) Confocal airy scan images of PVD with *ntuIs6* that expresses motor domains with rigor mutations, UNC-104-rigor::GFP and UNC-116-rigor::mCherry in wild type, *unc-33*, and *unc-33; Ex[des-2::UNC-33L]*. UNC-116- and UNC-104-associated MTs formed bundles, which extended into

dendrites and the axonal commissure (indicated by dotted box), while MT bundles in *unc-33* mutant were mislocalized to distal dendrites (yellow arrowheads). Brackets indicate ventral axon. Scale bar, 10 μm. (B) Quantification of UNC-116-and UNC-104-associated MT fluorescence intensity in the cell body. (C) Microtubule bundle width at anterior dendrite in (A) (short dotted line). (D) Line scan of fluorescence intensity of UNC-116- and UNC-104-associated MTs. Arrowhead indicates the peak intensity. Measured region is illustrated in the box. (E) Distance of the peak of MT intensity along anterior dendrite to the cell body. (F) Ratio of UNC-116- to UNC-104-associated MT intensity at axonal commissure (dotted box) and anterior proximal dendrite.(G) Confocal airy scan images of PVD expressing UNC-33L::internalGFP and UNC-116-rigor:: mCherry (*ntuEx179*). Arrowheads indicate UNC-33L puncta associated with UNC-116-rigor MT bundles. Brackets indicate UNC-33L puncta surrounding UNC-116-rigor MT bundles in the process. Scale bar, 5 μm. (H) Confocal airy scan images of PVD expressing UNC-33L:: internalGFP and tagRFP::UtrCH under *Pser2prom3* promoter (*ntuEx182*). Arrowheads indicate UNC-33L puncta co-localized with F-actin. a.u., arbitrary units of fluorescence intensity, Error bar represents SEM, n.s., not significant, *P<0.05; **P<0.01; ***p<0.001, one-way ANOVA. n = 20 (wild type), n = 19 (*unc-33*), n = 21 (*unc-33; Ex[des-2::UNC-33L]*).

UNC-33 puncta were surrounding microtubule bundles but colocalized with F-actin beneath the cell membrane (brackets in Fig 5G and 5H). The pattern is consistent with the idea that UNC-33 interacts with microtubule bundles to promote their high-order spindle-like organization under the cortical F-actin network.

## Mitochondria move along the stabilized MTs that requires CRMP/UNC-33

By imaging UNC-116-associated MT tracks and mitochondrial marker TOMM-20, we observed that mitochondria were placed along UNC-116-associated MT tracks (Fig 6A). Time-lapse movie showed that mitochondria moved along UNC-116-rigor-bound microtubule tracks (Fig 6B and S1 Video). We therefore wonder if mitochondria movement is abolished along defective microtubule tracks under *unc-33* mutant. In *ntuIs1* wild-type PVD, mitochondria in the cell body frequently underwent fusion and fission events. Long, filamentous mitochondria moved bi-directionally in dendrites, and some of them moved along axonal commissure to the axon (Fig 6C and S2 Video). However, in *unc-33* mutant, motile mitochondria were more frequently observed with small sizes and inconsistent movements (Fig 6D–6F and S3 Video), and these mitochondria rarely entered the axon (Fig 6G). Taken together, these results suggest that UNC-33 organizes an UNC-116-associated microtubule bundle structure that maintains the coherence of moving mitochondria and are essential for efficient axonal mitochondrial transport.

## Discussion

Neurons rely on active long-range transport to locate cytoplasmic components including vesicles and mitochondria to precise subcellular destinations. In this paper, we examined the role of motors in distributing mitochondria to different compartments in neuron, and we showed that transport regulators can control mitochondrial number specifically in different compartments based on both MT orientation and axon/dendrite identity. While TRAK-1 controls mitochondrial number in the MT-minus-end-out dendrite, JIP3/UNC-16 and CRMP/UNC-33 critically regulate axonal mitochondria. We further found that KIF5/UNC-116 is associated with a subset of MT bundles, which are organized into a spindle-like structure in the cell body. While most mitochondria localize closely to these bundles and move alone them, these KIF5-associated MT bundles likely function as main tracks for mitochondrial transport. We propose that CRMP/UNC-33 assembles these bundles to form a high-order MT circular network that supports coherent mitochondrial transport into the axon (Fig 7).

## UNC-33 organizes KIF5-associated microtubule bundles for axonal mitochondria

CRMP/UNC-33 possibly regulates MT in many aspects. Disruption of CRMP2 by expressing the dominant-negative form suppresses the formation of primary axon in cultured hippocampal neuron, while overexpression of CRMP2 induces multiple axon formation [19]. In *unc-33*

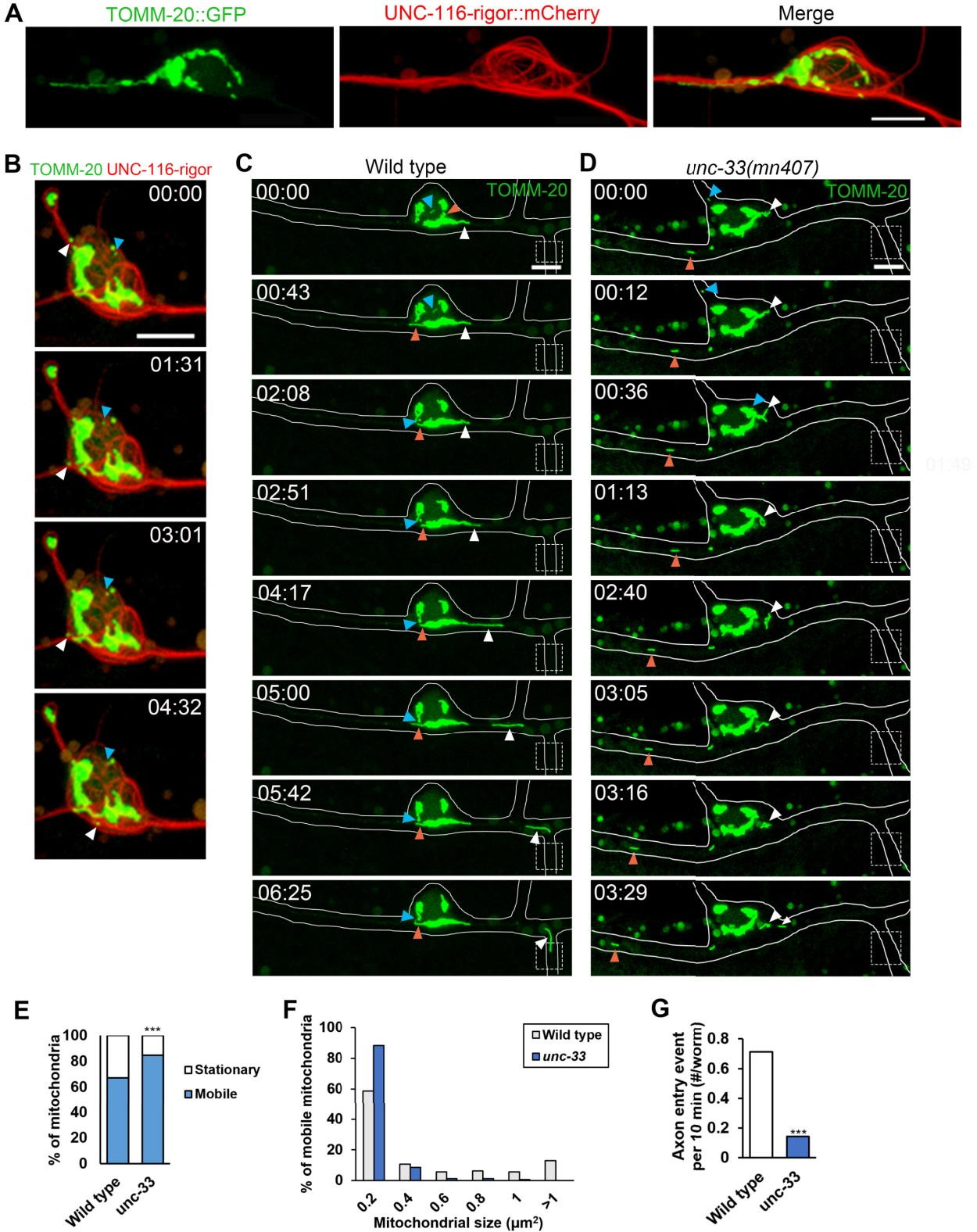

**Fig 6. UNC-33 organizes UNC-116-associated microtubule bundles for proper axonal mitochondrial transport.** (A and B) PVD neuron expressing mitochondrial marker TOMM-20::GFP and UNC-116-rigor::mCherry (*ntuIs10*). Mitochondria localized along UNC-116-associated MT

bundles (A). Time-lapse images showed that mobile mitochondria (arrowheads) moved on UNC-116-associated MT bundles (B). Arrowheads in distinct color indicate different moving mitochondria. (C and D) Time-lapse images of PVD mitochondria (*ntuIs1*) in wild type and *unc-33* mutant. Arrowheads in distinct color indicate different moving mitochondria. Dotted box indicates axonal commissure. Scale bar, 5 μm. (E) Percentage of mobile and stationary mitochondria in PVD. χ2 test, ***p<0.001. n = 263 (wild type), n = 192 (*unc-33*). (F) Distribution of mobile mitochondria in different sizes. n = 177 (wild type), n = 163 (*unc-33*). (G) Events of mitochondria entering the axonal commissure. Mitochondria over 0.2 μm$^2$ were scored in 10 minutes recording. Wild type: 21 worms, *unc-33*: 14 worms. χ2 test.

mutant, Maniar et. al. showed that synaptic cargoes and KIF1A/UNC-104 are mislocalized to dendritic regions in PVD, and dendritic microtubule polarity is disrupted, and Norris et. al. showed that UNC-33 influences MT plus-end distribution in VD growth cones *in vivo* [37,38]. Therefore, CRMP/UNC-33 is involved in axon specification and microtubule polarity. Whether the defect of mitochondrial axonal transport in *unc-33* mutant is due to the disruption of MT polarity in the axon? Studies of tracing the plus-end of MTs showed that axonal MTs in *unc-33* mutant remain normal with usual plus-end-out direction in the axon [26]. Although presynaptic components are reduced, we can still find around 30% of synaptic vesicle clusters present in *unc-33* axon, suggesting that a basal level of anterograde synaptic transport remains (Fig 4). Therefore, the nearly complete missing of axonal mitochondria in *unc-33* mutant is not likely caused by the disruption of MT polarity. CRMP-2 is proposed to mediate kinesin-1-dependent transport of tubulin and Sra-1/WAVE1 complex [34,39]. The C-terminal tail of CRMP-2 can directly bind kinesin light chain 1 (KLC1) *in vitro*, and thus it possibly regulates cargo recognition or motor activity of kinesin-1 [34]. Whether does CRMP/UNC-33 regulate mitochondrial transport by directly promoting kinesin-1 activity? Although the single mutant phenotypes of *unc-33* and *unc-116* were very similar, we found that the loss of *unc-33* severely exacerbated the mitochondrial transport defect of *unc-116* mutant (Fig 3). Furthermore, around the cell body, we observed more frequent mitochondrial movement in both anterograde and retrograde direction in *unc-33* mutant (Fig 6). Based on these findings, we infer that UNC-33/CRMP is not directly required for kinesin-1 motor activity in mitochondrial transport. In this study, we propose that UNC-33/CRMP is essential for a special subset of microtubule utilized by KIF5/UNC-116.

How does UNC-33 regulate MT organization or structure? CRMP2 monomer can bind tubulin heterodimers and promote microtubule assembly *in vitro* by its tubulin-interacting region in the N-terminal globular domain [18,21]. CRMP proteins usually form homo- or hetero-tetramers by their globular domain with the flexible C-terminal tail extending to its surface [21,40,41]. Meanwhile, another study showed that its C-terminal tail binds to and stabilizes microtubules *in vitro* and in cultured cells [20]. In addition to MT assembly and MT stabilization activity, incubation of recombinant CRMP2 protein with pre-formed paclitaxel-stabilized single microtubules prompts microtubule bundle formation [42]. Based on these studies, CRMP proteins may regulate MTs in three different levels: 1. CRMP monomer promotes tubulin incorporation into MT plus-ends. 2. CRMP tetramer acts as microtubule-associated protein that stabilizes MT by its C-terminal tail. 3. CRMP tetramer interacts with multiple MTs at the same time and therefore bundles MTs. Here, in *unc-33* mutant, we found that both tyrosinated dynamic MT and stable acetylated MTs (labelled by UNC-104-rigor and UNC-116-rigor respectively) generally decrease (Fig 5B), and the ratio of UNC-116-associated MTs (stable MTs) to UNC-104-associated MTs (dynamic MTs) reduces in axon (Fig 5F), consistent to the idea that UNC-33 affects MT assembly and stabilization. Moreover, while the width of MT bundle apparently decreases but MT bundles are still present in *unc-33* mutant (Fig 5C), the most dramatic effect is the missing of the spindle-like stable MT structure at the cell body (Fig 5A). Based on these findings, we infer that the primary role of UNC-33 in neuron is not only bundling MT but also organizing stable MT bundles into a high-order track system that

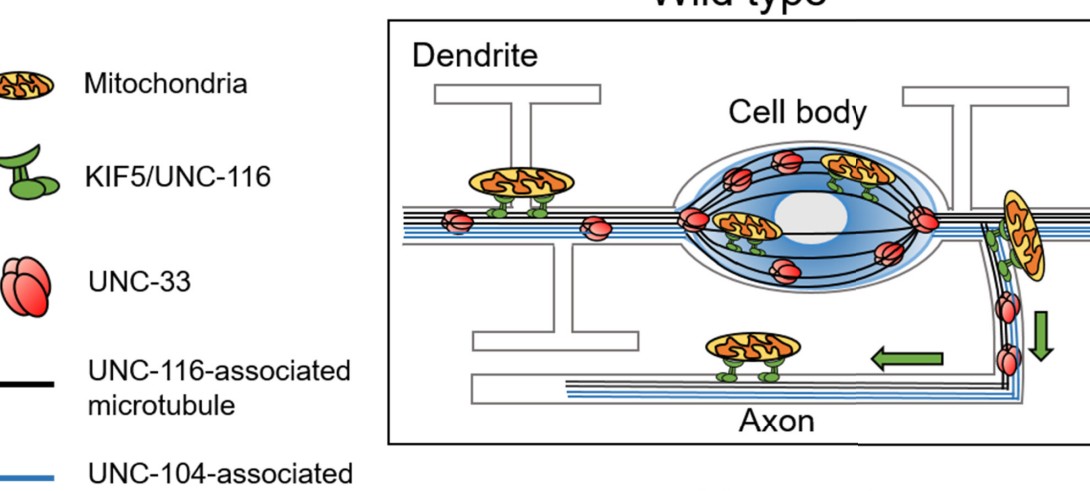

**Fig 7. A proposed model of CRMP/UNC-33 in organizing KIF5-associated microtubule bundles for axonal mitochondrial transport.** Neuron is a post-mitotic acentrosomal cell that required ordered MT tracks to form a network for efficient transport. While KIF5/UNC-116 selectively adopts stable MT bundles as its tracks, we proposed that CRMP/UNC33 organizes MT by pulling these stable MT bundles together to form a spindle-like high-order network at the cell body. Based on results of this study, axonal mitochondrial transport is mainly mediated by KIF5/UNC-116 motor, but suppressed by JIP3/UNC-16. In *unc-33* mutant, high-order spindle-like MT organization disappears, while large mitochondria fail to move and enter the axon.

supports highly efficient intracellular trafficking. Since MT bundles in *unc-33* mutant apparently disperse from the cell body, we propose that UNC-33 adheres stable MT bundles to some structure of cell body cortex, which is likely to be the F-actin network beneath the cell membrane (Fig 5H).

How CRMP/UNC-33 brings these MT bundles together at the correct subcellular location and supports the axonal tracks specifically is a very interesting question. A very recent study demonstrated that MTs in PVDs are anchored to the cell cortex by linking MT-associated CRMP/UNC-33 and the membrane-associated Ankyrin/UNC-44 via UNC-119 [36]. The membrane-associated complex acts as a pin to anchor MTs in a regular interval to the cell cortex, which is consistent to the working model in this study (Fig 7). Based on all these results, it is likely that CRMP/UNC-33 attaches stable MT bundles to each other and anchors these bundles to Ankyrin/UNC-44 for efficient traffic to neurites, especially the axon.

## Specific roles of different motors in mitochondrial transport

Mitochondrial transport in neuron is under the regulation of motor-based transport system, cytoskeleton structure, signaling pathway, and cellular activities like $Ca^{2+}$ level [8,35,43,44]. Through simple and efficient genetic approach, we explored mitochondrial transport regulation in the aspect of motor-transport system and microtubule organization. Since current studies mainly examine mitochondrial transport in cultured cells or in a segment of axons, we hence established a worm transgenic line to observe PVD mitochondrial distribution *in vivo*. PVD neurons have elaborate branched dendrites with two different MT polarity (Fig 1), so we will be able to distinguish the specific role of transport regulators in polarized trafficking and in distinct compartments, including the somato-dendritic and axonal regions. We systematically examine motors known to be involved in mitochondrial transport, including four plus-end motors, KIF5/UNC-116, KIF17/OSM-3, kinesin-3/KIF1/UNC-104, and kinesin-3 ortho-logue, KLP-6, and one minus-end motor, dynein. We determined the function of motors in three aspects, including the export of mitochondria from the cell body, distribution in dendritic regions with different MT polarity, and the axon or dendrite specificity. Our data reveal that the main driving force that dispatch mitochondria to neurites is generated by KIF5/UNC-116, which is responsible for exporting at least two-third of neurite mitochondria. In contrast, we found that dynein only mildly suppresses neurite mitochondrial numbers since they increase 11% in dynein mutant with no statistically significance (S1 Table), while other motors (KIF1/UNC-104, KIF17/OSM-3, and KLP-6) play no significant roles in controlling neurite mitochondrial number.

Apart from the regulation of total neurite mitochondrial numbers, axonal mitochondrial distribution is apparently regulated by these plus-end-directed motors in different ways, since the loss of KIF1/UNC-104 or KIF17/OSM-3 moderately increased axonal mitochondria, but KIF5/*unc-116* mutants had very few mitochondria left in the axon (Figs 1H, S1D and S1E). Based on these data, we infer that KIF5/UNC-116 is the main motor to export mitochondria from the cell body to neurites and also the main motor responsible for axonal distribution, while unexpectedly KIF1/UNC-104 and KIF17/OSM-3 suppress mitochondrial axonal distribution.

## The transport regulator TRAK-1 and JIP3/UNC-16 direct minus-end mitochondrial distribution

By modulating motor activity, motor adaptor plays important roles in mitochondrial transport. Based on our functional study of regulators, we surprisingly found a set of transport regulators specific for dendritic or axonal transport. Utilizing MT polarity distinct dendrites in PVD neurons, we were able to identify axonal or dendritic specific regulators by comparing the difference between the dendrite and the axon. The most characteristic motor adaptors required for mitochondrial transport are adaptor Milton and mitochondrial receptor Miro. Milton/TRAK1 tethers mitochondria through the outer membrane protein Miro, which is required for axonal mitochondrial distribution and promotes both anterograde and retrograde transports [10,45,46]. In our data, we observed that mitochondria were almost sequestered in the cell body in the absence of MIRO-1, which denotes the indispensible role of MIRO-1 in linking mitochondria to motor transport machinery (Fig 1C and 1F), consistent with the previous report [47]. In addition to MIRO-1, which acts as a necessary core component for mitochondrial transport, we have further identified distinct roles of three other regulators, including TRAK-1, UNC-16, and UNC-33.

TRAK proteins are regarded as main mitochondrial motor adaptors. In mammals, two TRAK proteins, TRAK1 and TRAK2, have different roles in mitochondrial transport. While

TRAK1 binds to both kinesin and dynein for axonal mitochondrial transport, TRAK2 is bound to dynein for dendritic transport [32]. In *C. elegans*, TRAK-1 is the only homolog of TRAK proteins, we unexpectedly found that 58% of mitochondria can move from the cell body to neurites in *trak-1* mutant when compared to *miro-1* that has no mitochondria present in neurites. Thus, we infer that TRAK-1 is not the only adaptor for mitochondria to attach motors. While KIF5/UNC-116 is the main motor mediating mitochondrial export to neurites, only 42% of the export events are associated with TRAK-1, and the rest of them are not. Interestingly, these mitochondria are distributed to posterior dendrite and the axon but not to anterior dendrite in *trak-1* mutant. Therefore, we conclude that the mitochondrial transport machinery fails in minus-end transport in the absence of TRAK-1, while it keeps half of its activity.

JIP3/UNC-16 can interact with kinesin-1 subunits and dynein light intermediate chain [15]. *jip3/unc-16* mutant axons have abnormal accumulated organelles, including lysosomes, endosomes, and mitochondria [48,49]. JIP3/UNC-16 has been proposed as an organelle gate-keeper at the axon initial segment [48]. These findings arise questions including what is the role of JIP3/UNC-16 in somatodendritic compartments, and whether JIP3/UNC-16 regulates mitochondrial distribution based on MT polarity or axon/dendrite identity. Here, we showed that total neurite mitochondria increased significantly in *unc-16*, and these mitochondria mainly distribute in MT-plus-end-out neurites (the posterior dendrite and the axon) (Fig 2B). Since the total neurite mitochondrial number is strictly regulated by KIF5/UNC-116, we infer that UNC-16 suppresses the activity of KIF5/UNC-116. Given that mitochondrial number significantly increases in both posterior dendrite and axon (S1 Table), we conclude that JIP3/UNC-16 not only regulates mitochondria in the axon but also in dendrites and the cell body.

## The motor KIF5/UNC-116 and the transport regulator CRMP/UNC-33 have specific roles in axonal mitochondrial distribution

In contrast to TRAK-1 and JIP3/UNC-16, we found that CRMP/UNC-33 is a specific and essential regulator for axonal mitochondria, since there were nearly no mitochondria in the axon of *unc-33* mutant, while plenty of mitochondria are still present in posterior dendritic segments (Fig 2C and 2E). Because MT polarity in these posterior dendrites are plus-end-out as axon, the defect of transport in one direction should have similar effects on posterior dendritic region and the axon. For example, we observed equal amount of mitochondria increase in comparing posterior dendritic region with axon in *dhc-1* mutants, indicating that dynein mediates minus-end transport equally in these two regions. However, similar to *unc-33* mutants, axonal mitochondria were much more severely affected than those in posterior dendritic region in *unc-116* mutants. Taken together, these results reveal that axonal mitochondrial distribution is not only regulated by microtubule polarity but also required specific regulators and transport machineries including CRMP/UNC-33 and KIF5/UNC-116, while minus-end mitochondrial distribution is directed by UNC-16 and TRAK-1.

## Conclusion

In neuron, although mitochondria distribute in all subcellular compartments including the cell body, dendrites, and the axon, we found that each compartment has specific regulatory mechanisms for mitochondrial transport. In the cell body, the export of mitochondria is mediated by KIF5/UNC-116 and MIRO-1, while partially requires TRAK-1, but is suppressed by JIP3/UNC-16. In dendrites, mitochondrial distribution requires dynein and also TRAK-1. For axonal mitochondria, KIF5/UNC-116 and CRMP/UNC-33 are necessary. While neurons are post-mitotic cells with no centrosome as MT-organization center, we surprisingly found that

CRMP/UNC-33 is required for organizing a high-order MT structure in the cell body and selectively supports stable MT in the axon.

## Material and methods

### Strains and genetics

Worms were raised on NGM plates seeded with *Escherichia coli* OP50 at 20˚C. The Bristol N2 strain was used as wild type. Mutant alleles used in this study were: *LG I*: *dhc-1(or195)*, *trak-1 (tm1572)*, *nud-2(ok949)*; *LG II*: *unc-104(e1265)*; *LG III*: *unc-116(e2310)*, *klp-6(my8)*, *unc-16 (e109)*; *LG VI*: *miro-1(tm1966)*, *osm-3(p802)*, *unc-33(mn407)*, *bicd-1(ok2731)*, *dnc-1(or404)*. The temperature sensitive strains *dhc-1(or195)* and *dnc-1(or404)* were maintained at 15˚C, and shifted to 20˚C for phenotype analysis. Standard PCR was performed for strain genotyping.

Integrated transgenes used were: *ntuIs1[Pdes-2::TOMM-20::GFP 20 ng/μl; Pdes-2::myr:: mCherry 80 ng/μl; Podr-1::GFP 40 ng/μl]X, ntuIs2[Pdes-2::UNC-116::GFP 20 ng/μl; Pdes-2:: myr::mCherry 80 ng/μl; Pmyo-2::GFP 5 ng/μl]III, ntuIs6[Pdes-2::UNC-116(G237A)::mCherry 10 ng/μl; Pdes-2::UNC-104(E250K)::GFP 10 ng/μl; Podr-1::GFP 40 ng/μl]III, ntuIs10[Pdes-2:: TOMM-20::GFP 20 ng/μl; Pdes-2::UNC-116(G237A)::mCherry 10 ng/μl; Podr-1::GFP 40 ng/μl]*. Extrachromosomal lines used were: *ntuEx22, 23[Pdes-2::UNC-33L 10 ng/μl; Pmyo-2::GFP 5 ng/ μl], ntuEx24-26[Pdes-2::UNC-33M 10 ng/μl; Pmyo-2::GFP 5 ng/μl], ntuEx27-30, 126[Pdes-2:: UNC-33S 10 ng/μl; Pmyo-2::GFP 5 ng/μl], ntuEx35[Pdat-1::myr::mCherry 20 ng/μl; Pdat-1:: TOMM-20::GFP 10 ng/μl; Podr-1::GFP 40 ng/μl], ntuEx178[Pmec-4::myr::mCherry 20 ng/μl; Pmec-4::TOMM-20::GFP 15 ng/μl; Podr-1::GFP 40 ng/μl], ntuEx179[Pdes-2::UNC-33L::internalGFP 20 ng/μl; Pdes-2::UNC-116(G237A)::mCherry 10 ng/μl; Podr-1::GFP 40 ng/μl], ntuEx182[Pdes-2::UNC-33L::internalGFP 20 ng/μl; Pser2prom3::tagRFP::UtrCH 20 ng/μl* [50]; *Podr-1::GFP 40 ng/μl], ntuEx185[Pmec-4::UNC-116(G237A)::mCherry 10 ng/μl; Pmec-4::UNC-104(E250K)::GFP 10 ng/μl; Podr-1::GFP 40 ng/μl], wyEx5216[Pdes-2::GFP::RAB-3; Pser2-prom3::myr::mCherry; Podr-1::RFP]* [51]. *Podr-1::GFP* and *Pmyo-2::GFP* were used as co-injection markers.

### Constructs and transgenic worms

Expression clones were made in the pSM vector, a derivative of pPD49.26 (A. Fire) with extra cloning sites (*S. McCarroll* and *C. I. Bargmann*, personal communication). The *Pdes-2* promoter was used for PVD-specific expression, and *Pmec-4* promoter was used for touch receptor neuron expression. *Pdes-2* promoter fragment was amplified as previously described [37], and *Pmec-4* promoter fragment was amplified as previously described [52], both with *Sph*I and *Asc*I sites (Phusion, NEB). cDNAs of the fragment of the first 54 amino acids of *tomm-20* were amplified with *Nhe*I and *Kpn*I sites [53], and *unc-33*(long, medium, and short forms) with *Asc*I and *Nhe*I sites from cDNA library. To visualize PVD neuronal morphology and mitochondria, we generated *Pdes-2::TOMM-20::GFP* (pHW2*)* and *Pdes-2::myr::mCherry* (pHW5). Myristoylation motif was amplified with *Nhe*I and *Kpn*I sites to construct *Pdes-2::myr::mCherry* (pHW5) as previously described [54]. *ntuIs1* was generated by integrating extra chromosomal array containing pHW2 and pHW5 into N2 genome by UV irradiation at 100 μJ/cm$^2$ [55]. *ntuIs1* was outcrossed at least six times and was crossed to different mutant strains for phenotype analysis. To visualize mitochondria of ALM, *Pmec-4::myr::mCherry* (pFY1) and *Pmec-4:: tomm-20::GFP* (pYC28) were constructed and co-injected into N2 worm to get *ntuEx178*. To visualize PDE neuron, *Pdat-1* promoter was constructed as previously described [56].

For *unc-33* rescue experiment, *UNC-33L*, *UNC-33M*, or *UNC-33S* cDNAs were constructed with *Pdes-2* as pYC1, pYC2, pYC3, which were injected separately into N2 worm to get

*ntuEx22*, *23* with pYC1, *ntuEx24-26* with pYC2, and *ntuEx27-30*, and *ntuEx126* with pYC3. P*des-2*::UNC-116::*GFP* (pHH1) was constructed with P*des-2* promoter fragment into *Pitr-1*::UNC-116::*GFP* (pCO25). To generate UNC-116 and UNC-104 rigor mutant form for labeling their associated microtubules, motor rigor mutant UNC-116(G237A) and UNC-104(E250K) were generated by PCR from cDNA library and inserted into pHH1 as P*des-2*::UNC-116(G237A)::*mCherry* (pCH9) and P*des-2*::UNC-104(E205K)::*GFP* (pCH4). The motor rigor mutation sites are at the ATP-binding motif for motors to bind MTs without dissociation as described [25,57]. To express functional UNC-33L::internalGFP, we inserted GFP into *unc-33* gene at the start of S isoform to generate pYC26 as previously described [36]. To visualize F-actin in PVD, we expressed P*ser2prom3*::*tag*::*RFP*::*UtrCH*, which is a gift form Hannes E Bűlow lab.

## Image acquisition

Fluorescence images of live *C. elegans* were acquired using Zeiss Axio Imager microscope (Carl Zeiss, Germany) with X-cite light source and 63x/1.4NA oil objective and Evolve 512 EMCCD camera (photometrics) at 23°C for quantification. Worms at mid-L4 stage were immobilized with 2–3 μl of 5 mM levamisole (Sigma-Aldrich) on 3% agarose pads. For confocal images in Figs 1–4 and S1–S3, worms were immobilized with 10 mM levamisole on 10% agarose pads and imaged by a Zeiss LSM710 microscope (Carl Zeiss, Germany) with 63x/1.4NA oil objective and 488 nm and 561 nm laser lines. For each worm, 4–5 images were stitched together to cover the whole body (Adobe Photoshop). Worm images were straightened and measured by ImageJ. High-resolution UNC-116 and UNC-104 motor-rigor fluorescence images were captured by the Airyscan mode (SR mode) of Zeiss LSM880 confocal microscope. Mitochondrial dynamics was recorded by the Airyscan mode (R/S mode) of LSM880 at 30–40 seconds per frame for 10 minutes. Line scan graphs were generated by ZEN 2.6 lite software.

## Quantification and statistical analysis

For mitochondrial distribution, 20 worms of each genotype at mid-L4 stage were imaged and scored, unless stated otherwise. Quantification of mitochondrial distribution of each genotype was summarized in S1 Table. Based on these distribution data, axonal mitochondrial percentage and dendritic distribution pattern were analyzed (Figs 1–3). One-way ANOVA, Student's t test, and Chi-square test ($\chi^2$) were performed for statistics. Diagrams were made by Excel 2016 and GraphPad Prism 6.0 software.

## Supporting information

**S1 Fig. Motors that regulate PVD mitochondrial distribution and morphology.** (A) Mitochondria in PVD cell body of wild type, *miro-1*, *unc-116*, and *unc-33* mutants. Scale bar, 5 μm. (B) Quantification of the TOMM-20::GFP intensity in the cell body. n = 21 (wild type), n = 25 (*miro-1*), n = 21 (*unc-116*), n = 25 (*unc-33*). (C) Quantification of the total TOMM-20::GFP intensity in all neurites and the cell body. The intensity was normalized to the body volume for each genotype. n = 10 for each genotype. (D-F) PVD neurons in *unc-104(e1265)*, *osm-3(p802)*, *klp-6(my8)* mutants. Arrowheads indicate mitochondria. Scale bar, 20 μm. G) Mitochondria in PVD cell body of wild type and *klp-6* mutant. Arrowheads denote aggregated mitochondria. Scale bar, 5 μm. a.u., arbitrary units of fluorescence intensity. Error bar represents SEM, n.s., not significant, *P<0.05; **P<0.01; ***p<0.001, compared to wild type, one-way ANOVA. (EPS)

**S2 Fig. UNC-33 regulates the distribution of mitochondria in ALM and PDE neurons.** (A and B) ALM neuron expressing myr-mCherry (red) and mitochondrial marker TOMM-20:: GFP (green) under *Pmec-4* promoter (*ntuEx178*) of wild type (A), and *unc-33(mn407)* (B). Mitochondria are indicated by arrowheads. Dotted box indicates ventral nerve ring branch. (A' and B') Magnified view of the corresponding region of the nerve ring branch of wild type (A') and *unc-33* (B'). Brackets indicate the nerve ring branch. While ALM neuron is labeled by mCherry (red), green olfactory neurons are also present due to the co-injection marker *Podr-1::gfp*. Scale bar, 10 μm. (C) Quantification of the number of mitochondria in anterior neurite in L4 wild type and *unc-33* mutant. Error bar represents SEM, T-test, ***$p<0.001$, n = 17 (wild type), n = 16 *(unc-33)*. (D and E) Image and schematic of PDE neuron expressing myr-mCherry (red) and mitochondrial marker TOMM-20::GFP (green) under *Pdat-1* promoter (*ntuEx35*) of wild type (D), and *unc-33(mn407)* (E). Scale bar, 10 μm.
(EPS)

**S3 Fig. Expression of UNC-33L in PVD neuron rescues morphological and mitochondrial transport defects of *unc-33* mutant.** (A) Diagram of three protein isoforms (long, medium, short) encoded by *unc-33* with their putative domains. The N-terminal region of UNC-33L and UNC-33M are marked with grey color. Blue box indicates tubulin-binding domain (TBD), and yellow box indicates microtubule-binding domain (MT). Transgenes of each isoform are expressed in *unc-33* mutant PVD neurons to test the rescue activity. "+" represents both *unc-33* morphology and axonal mitochondrial defects can be rescued. "-"represents no rescue effect. (B) Defective PVD morphology and disrupted mitochondrial transport in *unc-33* are cell-autonomously rescued by expressing UNC-33L. Arrowheads indicate mitochondria. Scale bar, 20 μm. (C) Quantification of the percentage of the worms with rescued axonal mitochondrial number in *unc-33* mutant. ***$p<0.001$, χ2 test. Number of scored worms were indicated in the graph.
(EPS)

**S4 Fig. UNC-33 organizes microtubule structure in touch receptor neurons.** Confocal airy scan images of ALM (A), and PLM (B) expressing UNC-116-rigor::mCherry and UNC-104-rigor::GFP under *Pmec-4* promoter (*ntuEx185*) of wild type and *unc-33* mutants. Scale bar, 5 μm.
(EPS)

**S1 Table. Mitochondrial distribution in different mutant strains in every section of PVD neuron.**
(TIF)

**S1 Video. Mitochondria movement along UNC-116-associated MT tracks.**
(MP4)

**S2 Video. Time-lapse movement of mitochondria in the PVD neuron in wild type.**
(MP4)

**S3 Video. Time-lapse movement of mitochondria in the PVD neuron in *unc-33*.**
(MP4)

## Acknowledgments

We thank Hsun Li, Chien lab, Academia Sinica and imaging core facility of the First Core Labs, National Taiwan University for technical support. We thank Cheng-Ting Chien, Academia Sinica, and Jing-Jer Lin, Hsin-Yue Tsai, and Shu-Yi Huang, National Taiwan University

for comments on the manuscript. We thank John Wang, Academia Sinica and Shih-Peng Chan, National Taiwan University for genomic single-copy insertion technique. We thank the *C. elegans* Core Facility of the National Core Facility for Biopharmaceuticals, Ministry of Science and Technology. We thank the Caenorhabditis Genetics Center and National BioResource Project for providing mutant strains. We thank WormBase for providing informative data and sequences.

## Author Contributions

**Conceptualization:** Ying-Chun Chen, Chan-Yen Ou.

**Data curation:** Ying-Chun Chen.

**Formal analysis:** Ying-Chun Chen, Hao-Ru Huang.

**Funding acquisition:** Chan-Yen Ou.

**Investigation:** Ying-Chun Chen.

**Methodology:** Hao-Ru Huang, Chia-Hao Hsu.

**Supervision:** Chan-Yen Ou.

**Writing – original draft:** Ying-Chun Chen, Chan-Yen Ou.

**Writing – review & editing:** Ying-Chun Chen, Hao-Ru Huang, Chan-Yen Ou.

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
