## [Decision Letter · Decision Letter 0]

15 Aug 2020

Dear Dr Ou,

Thank you very much for submitting your Research Article entitled 'CRMP/UNC-33 organizes microtubule bundles for KIF5-mediated mitochondrial distribution to axon' to PLOS Genetics. Your manuscript was fully evaluated at the editorial level and by independent peer reviewers. The reviewers appreciated the attention to an important problem, but raised some substantial concerns about the current manuscript. Based on the reviews, we will not be able to accept this version of the manuscript, but we would be willing to review again a much-revised version. We cannot, of course, promise publication at that time.

As you will see, reviewers have mixed opinions as to whether the advance described in the present manuscript is sufficient for this journal. As such, additional experimental work along the lines suggested by reviewers (e.g. examination of other neuron types; analysis of MT polarity) would be required for a revised manuscript to be considered. The editors note that Reviewer #3 criticized the resolution of the figures in the PDF, but we also note that you had indeed provided high quality TIFF images with your submission which were available to the reviewers on our online system - we will make sure the reviewer is aware of these if further external review is necessary.

If you decide to revise the manuscript for further consideration at PLOS Genetics, please aim to resubmit within the next 60 days, unless it will take extra time to address the concerns of the reviewers, in which case we would appreciate an expected resubmission date by email to plosgenetics@plos.org.

[LINK]

We are sorry that we cannot be more positive about your manuscript at this stage. Please do not hesitate to contact us if you have any concerns or questions.

Yours sincerely,

Andrew D. Chisholm

Associate Editor

PLOS Genetics

Gregory P. Copenhaver

Editor-in-Chief

PLOS Genetics

Reviewer's Responses to Questions

**Comments to the Authors:**

Reviewer #1: The manuscript by Chan-Yen Ou group entitled “CRMP/UNC-33 organizes microtubule bundles for KIF5-mediated mitochondrial distribution to axon” evaluated how mictochondrial distribution in different compartments of neurons is regulated. They examined mitochondrial motors and transport regulators for their roles in mitochondrial transport using the PVD neuron as a model. This is an important topic, given that mitochondrial trafficking is essential for neuronal function and its dysregulation is involved in various neurodegenerative diseases.

Despite the characterization of factors regulating mitochondrial transport, how is this regulated in different cell compartments is less clear. The authors established a transgenic reporter for mitochondrial distribution in PVD neurons and examined how mitochondrial distribution was affected by loss of function of kinesins (unc-116, unc-104, klp-6 and osm-3), dynein (dhc-1), and transport regulators (trak-1, unc-16 and unc-33). They revealed distinctive roles of these genes in mitochondrial transport and analyzed how they genetically interact. As unc-33 is known to regulate microtubule organization, they went ahead to test whether unc-33 impact motor polarized mitochondrial distribution by regulating general cytoskeletal arrangement. They concluded that UNC-33 selectively stabilizes axonal microtubules to promote UNC-116-mediated mitochondrial transport to axons. Overall, I found the genetics experiments to be quite compelling. However, I believe a few issues will need to be addressed.

Major concerns:

1. The authors established a very nice genetic reporter to study mitochondrial transport in PVD neurons and identified specific roles of the motors and regulators in different compartments of the neuron. Are these findings neuron type-specific, or can these findings be applied to other neuron types, such as touch sensory neurons and DA9 neuron, which have long been used to study vesicular trafficking and microtubule organization?

2. The authors have concluded that “UNC-33 sticks MT bundles together to form a spindle-like high-order structure in the cell body and selectively maintains stabilized MTs for KIF5”. They did not provide images showing UNC-33 and UNC-116-rigor-GFP, but based on the Mossci reporter of UNC-33::GFP, the subcellular localization pattern of UNC-33 does not match with UNC-116-rigor-GFP pattern (i.e. spindle-like pattern in soma).

3. They propose that UNC-33 adheres stable MT bundles to some structure of cell body cortex, which is likely to be the F-actin network beneath the cell membrane. Can they test this?

4. They have shown that UNC-33 is localized in soma, axon and dendrites, then why UNC-33 selectively stabilize microtubules in axons? They discussed that this might not be due to microtubule polarity. Testing this in a different neuron type might help too.

5. Can microtubule targeting agents rescue the defects in unc-33 mutant?

Minor issue:

The figures provided for review are in extremely low resolution, making it super hard to read and evaluate the data.

Typo in line 342-343: “Whether dose CRMP/UNC-33 regulate mitochondrial transport by directly promoting kinesin-1 activity?”

Reviewer #2: The manuscript by Ying-Chun Chen et al addressed how mitochondria is transported differentially in axon and dendrites using PVD neuron model in worm. They used mitochondrial reporter TOMM20::GFP and mutants deleting microtubule-based motors and their regulatory subunits to address the above-mentioned question. While it is really appreciated that PVD neuron would be a good system to look at the mitochondrial transport in neuron and authors develop a nice system to study this area. Using this system, the authors showed that KIF5/UNC-116 and dynein/DHC-1 is important for transport of mitochondria in axon and dendrites respectively. The mitochondrial adaptor Miro-1 plays an important role in mitochondrial distribution in all the neurites. According to the authors, more intriguing is the finding that CRMP family protein UNC-33 plays an important role in this process by regulating the organization of microtubule cytoskeleton in axon and dendrites. I completely agree that it is a nice characterization of this new system for studying mitochondrial transport. However, a major concern is that lack of new insight this study gives on mitochondrial distribution in neuronal cell.

1) For example, the role of KIF5/UNC-116 (Campbell et al 2014 J Neurosc, Han et al 2016 Neuron), Miro-1 (Han et al 2016 Neuron and many other studies) and dynein (van Spronsen M et al 2013 Neuron) is well established in axonal and dendritic transport of mitochondria. There are many other reports on the role of kinesin-1 and dynein in axonal transport of mitochondria.

2) The major finding that UNC-33 is important for mitochondrial transport lacks sufficient insight. For example, ‘Maniar et al 2011 Nat Neurosc’ clearly demonstrated that UNC-33 is important for establishment of microtubule polarity in axon and dendrites of PVD neuron. In the absence of the same MTs are disorganized and affects distribution of motors and cargoes in synapse and dendrites. Apart from that also other studies that highlight the importance of UNC-33/CRMP in neuronal microtubule organization in C. elegans (Martin Harterink et al 2018 JCS) and vertebrate model. Essentially, it’s important that authors highlight their findings that advances this field. On that basis, I feel that this study would be more suitable for elsewhere.

3) The authors may highlight how UNC-33 might be specifically important for mitochondrial transport? Can they provide some evidence by disturbing the specific function of UNC-33 related to mitochondria that only mitochondrial transport is affected without altering the distribution of other components such as RAB-3.

4) The UNC-33L isoform was shown to regulate microtubule organization earlier (Maniar et al 2011). Although authors suggest M and S isoform does not rescue, no quantitative data is presented.

5) In figure-5, the authors claim that UNC-33 is involved in organization of UNC-116-bound microtubule tracks. How can they be sure that the other microtubule population is unaffected?

6) I suggest the authors that they proofread the manuscript for the language and grammatical errors. At times, it’s difficult to follow. It needs a major rewriting.

Reviewer #3: This work describes the roles of dynein and kinesin MT motor proteins, adapters, and the MT organizing molecule UNC-33/CRMP in mitochondrial distribution in neurons. The PVD sensory neuron has process with MTs of distinct polarities: + end out axon and posterior dendrite; and - end out anterior dendrite. They find that specific motor proteins and adapters are responsible for mitochondrial distribution in these processes. They also find that UNC-33/CRMP is required to establish a unique MT structure in the cell body utilized by UNC-116/KIF5 for mitochondrial entry into processes from the cell body. The mitochondrial distribution assay in the PVD neuron is nice, and is the basis of many experiments in the manuscript. The live imaging of mitochondrial entry into processes was good. These are interesting and significant results. The experiments are in general well designed and controlled, with appropriate statistical analyses. The figures, however, were of too low a resolution to assess some of the claims made by the authors in the text. My comments to improve the manuscript are below.

1) The resolution of the figures in the pdf copy that I downloaded was really low, making it difficult in some instances to see in micrographs what the authors describe in the text. Likewise, some of the graphs and data figures were small and at this resolution, difficult to see clearly.

2) Many of the experiments rely on quantifying mitochondrial prevalence in distinct processes and sub-sections of processes. The authors make claims about increased or decreased mitochondrial movement. However, it is also possible that more or fewer overall mitochondria are being generated. This should be addressed in each experiment.

3) Add a reference for the unc-33(mn407) deletion allele or add a figure with the deletion data.

4) The argument about unc-16/JIP3 gating rather than minus end specificity would be bolstered by analyzing total mitochondria in this mutant.

5) The authors find that mutations in the dhc-3 dynein heavy chain partially suppress defects in unc-33 mutants. Because suppression is not complete, the authors argue that dhc-3 and unc-33 act independently and that unc-33 is unlikely to regulate mitochondrial transport via dhc-3. Is the suppression significant compared to unc-33 alone? If so, I would argue that this suggests that they do interact, and that dhc-3 might be one of multiple unc-33 partners in mitochondrial transport.

6) Defects in the unc-33; unc-116 double mutant are increased relative to single mutants. Is this a synergistic effect, or additive? If it is synergistic (i.e. greater than the predicted additive effect of the mutations alone), this suggests parallel, converging pathways. If it is additive, this suggests mechanistically independent pathways. This should be addressed, as it gets at mechanism.

7) The authors mention that MT polarity is disrupted in some mutants. This seems like an important experiment to perform for all mutants, but especially for unc-33. Do unc-33 mutants have MT polarity defects?

8) The Results section entitled “UNC-33 organizes microtubule bundles in PVD cell body and stabilizes microtubules in axon” is written in an unclear manner. Part of this is due to referring to MTs using distinct terminology. The authors switch between acetylated versus tyrosinated MTs, stable versus dynamic MTs, and those labeled by the unc-116rigor and unc104rigor. I would suggest described the reagents and what they selectively label, and then describe experiments by using the reagent name. For example, instead of

“In the absence of UNC-33, the spindle-like acetylated MT bundles were severely diminished in the cell body, and the general level of tyrosinated MTs in the cortical region of the cell body was also decreased, but the circular mesh of tyrosinated MTs was still present around the nucleus (Fig 5A and 5B).”

Use this terminology:

“In the absence of UNC-33, the spindle-like UNC-116rigor signal was severely diminished in the cell body, and the general level of UNC-104rigor signal in the cortical region of the cell body was also decreased, but the circular mesh of unc-104rigor was still present around the nucleus (Fig 5A and 5B).”

9) A reference for the rigor technique is missing.

10) Given the resolution of the figures, it was impossible for me to see the spindle-shaped UNC-116rigor structure or the circular mesh of unc-104rigor in Figure 5. Because of this, it was impossible for me to assess the effect of unc-33 on these structures.

11) Work from my lab has shown that unc-33 influences MT + end distribution in VD growth cones in vivo (Norris et al., 2014). The results presented here are consistent with this result. I would ask the authors to consider referencing this work in discussions about known roles of unc-33/CRMP.

**Have all data underlying the figures and results presented in the manuscript been provided?**

Reviewer #1: Yes

Reviewer #2: Yes

Reviewer #3: Yes

PLOS authors have the option to publish the peer review history of their article (what does this mean?). If published, this will include your full peer review and any attached files.

Reviewer #1: No

Reviewer #2: No

Reviewer #3: **Yes: **Erik A. Lundquist

---

## [Decision Letter · Decision Letter 1]

11 Nov 2020

Dear Dr Ou,

We are pleased to inform you that your manuscript entitled "CRMP/UNC-33 organizes microtubule bundles for KIF5-mediated mitochondrial distribution to axon" has been editorially accepted for publication in PLOS Genetics. Congratulations!

Yours sincerely,

Andrew D. Chisholm

Associate Editor

PLOS Genetics

Gregory P. Copenhaver

Editor-in-Chief

PLOS Genetics

Comments from the reviewers (if applicable):

Reviewer's Responses to Questions

**Comments to the Authors:**

Reviewer #1: The revised manuscript has fully addressed my comments raised previously. Therefore, I support its publication.

Reviewer #2: The authors have appropriately addressed the comments made by the reviewers. Although, I had some reservations on the novelty of this finding as the role of UNC-33 in microtubule organization is well established, I feel that this work very nicely establishes PVD neuron of worm as a model to study mitochondrial transport and see the consequences in various conditions. The authors have done a through characterization of the transport system in this work and therefore it should be published.

Reviewer #3: The authors responded appropriately to reveiwer comments. The additional experiments added, including total mitochondrial levels, strengthen the manuscript.

**Have all data underlying the figures and results presented in the manuscript been provided?**

Reviewer #1: Yes

Reviewer #2: Yes

Reviewer #3: Yes

PLOS authors have the option to publish the peer review history of their article (what does this mean?). If published, this will include your full peer review and any attached files.

Reviewer #1: No

Reviewer #2: No

Reviewer #3: **Yes: **Erik A. Lundquist

**Data Deposition**

http://datadryad.org/submit?journalID=pgenetics&manu=PGENETICS-D-20-01044R1

**Press Queries**

---

## [Editor Report · Acceptance letter]

5 Feb 2021

PGENETICS-D-20-01044R1 

CRMP/UNC-33 organizes microtubule bundles for KIF5-mediated mitochondrial distribution to axon 

Dear Dr Ou, 

We are pleased to inform you that your manuscript entitled "CRMP/UNC-33 organizes microtubule bundles for KIF5-mediated mitochondrial distribution to axon" has been formally accepted for publication in PLOS Genetics! Your manuscript is now with our production department and you will be notified of the publication date in due course.

With kind regards,

Alice Ellingham

PLOS Genetics

On behalf of:
